

# Fractal Scaling Analysis of Groundwater Dynamics in Confined Aquifers

Tongbi Tu[1], Ali Ercan[1, 2], M. Levent Kavvas[1, 2]

[1]J.Amorocho Hydraulics Laboratory, Dept. of Civil and Environmental Engineering, University of California, Davis, CA 95616, USA

[2]Hydrologic Research Laboratory, Dept. of Civil and Environmental Engineering, University of California, Davis, CA 95616, USA

*Correspondence to*: Ali Ercan (aercan@ucdavis.edu)

**Abstract.** Groundwater closely interacts with surface water and even climate systems in most hydro-climatic settings. Fractal scaling analysis of groundwater dynamics is of significance in modeling hydrological processes by considering potential temporal long-range dependence and scaling crossovers in the groundwater level fluctuations. In this study, it is demonstrated that the groundwater level fluctuations of confined aquifer wells with long observations exhibit site-specific fractal scaling behavior. Detrended fluctuation analysis (DFA) was utilized to quantify the monofractality; and Multifractal detrended fluctuation analysis (MF-DFA) and Multiscale Multifractal Analysis (MMA) were employed to examine the multifractal behavior. The DFA results indicated that fractals exist in groundwater level time series, and it was shown that the estimated Hurst exponent is closely dependent on the length and specific time interval of the time series. The MF-DFA and MMA analyses showed that different levels of multifractality exist, which may be partially due to a broad probability density distribution with infinite moments. Furthermore, it is demonstrated that the underlying distribution of groundwater level fluctuations exhibits either non-Gaussian characteristics which may be fitted by the Lévy stable distribution or Gaussian characteristics depending on the site characteristics. However, fractional Brownian motion (fBm), which has been identified as an appropriate model to characterize groundwater level fluctuation is Gaussian with finite moments. Therefore, fBm may be inadequate for the description of physical processes with infinite moments, such as the groundwater level fluctuations in this study. It is concluded that there is a need for generalized governing equations of groundwater flow processes, which can model both the long-memory behavior as well as the Brownian finite-memory behavior.

## 1 Introduction

Groundwater in both confined and unconfined aquifers is usually a complex and dynamic system which highly interacts with surface water and even climate systems in most hydro-climatic settlings, due to its discharge to rivers and streams, and its recharge that is affected by various related physical processes, such as precipitation, evapotranspiration and infiltration (Green et al., 2011; Joelson et al., 2016; Li and Zhang, 2007; Rakhshandehroo and Amiri, 2012; Taylor et al., 2013). These processes, which take place over various spatiotemporal scales, add further complexity to groundwater systems. Groundwater level fluctuations



dynamically reflect the responses of an aquifer to its diverse inputs and outputs. Consequently, groundwater level fluctuations are often non-stationary, rendering variabilities over different spatial and temporal scales and resulting in no dependence on single representative spatial and temporal scales. Therefore, groundwater level fluctuations are often characterized as scale-free processes and modeled as fractional Brownian motion

(Hardstone et al., 2012; Yu et al., 2016). Not necessarily totally random, groundwater level fluctuations may demonstrate long-range dependence through time, implying a power-law relationship over a variety of time scales, which can be represented by fractals (Yu et al., 2016).

Fractal analysis of both persistent and anti-persistent behavior has been extensively utilized to investigate possible relationships in variability among various scales (Blöschl and Sivapalan, 1995). Temporal fractal

scaling analysis of groundwater dynamics can be essential to a better understanding of the modeling of hydrological processes by considering the temporal correlations and scaling cascading issues, since groundwater closely links to surface water in hydrological modeling and hydrological models are built upon certain temporal and spatial scales (Blöschl and Sivapalan, 1995; Yu et al., 2016). In fact, groundwater dynamics was found to provide a positive feedback to the memory of land surface hydrological processes in

the climate systems (Lo and Famiglietti, 2010). Furthermore, fractal analysis of groundwater level fluctuation may shed light on the modeling of flow and solute transport in porous media (Kavvas et al., 2017; Tu et al., 2017), as non-Fickian transport behavior has been found in many laboratory experiments and field studies (Berkowitz and Scher, 2009; Gjetvaj et al., 2015; Kang et al., 2015).

Detrended Fluctuation Analysis (DFA), originally used to analyze long-range power-law correlations (i.e.,

persistent fractal scaling behavior) of time series, is considered a more powerful method to quantify the scaling parameter or the Hurst exponent for its capacity in detecting nonstationarities and distinguishing seasonal oscillations from intrinsic fluctuations, compared with conventional methods, such as R/S analysis or the variation method (Dubuc et al., 1989; Hardstone et al., 2012; Shang and Kamae, 2005). In order to characterize multifractal structures within complex nonlinear heterogeneous processes, Multifractal

Detrended Fluctuation Analysis (MF-DFA(Kantelhardt et al., 2002) was developed on the framework of DFA, which is mostly used to quantify monofractality. DFA and MF-DFA have been widely applied to evaluate fractal scaling properties of rainfall and streamflow time series in hydrology (Kantelhardt et al., 2002; Koscielny-Bunde et al., 2006; Labat et al., 2011; Livina et al., 2003; Matsoukas et al., 2000; Zhang et al., 2008).

More specifically, in subsurface hydrology DFA was first adopted by Li and Zhang (2007) to systematically evaluate fractal dynamics of groundwater systems. They analyzed four years of continuous hourly data from seven wells and found groundwater level fluctuations are likely to follow fractional Brownian motion (fBm), and temproal scaling crossovers exist in the fluctuations. These findings were later confirmed by Little and Bloomfield (2010), Rakhshandehroo and Amiri (2012) and Yu et al. (2016) with the application of DFA on

hourly or in 15-minute interval data for up to 5 years from 7 wells, daily data for 6 years from 2 wells, and daily data from 22 wells that have more than 2,500 records, respectively. Rakhshandehroo and Amiri (2012)



further utilized MF-DFA to evaluate the multifractality of groundwater level fluctuations and concluded the extent of multifractality in groundwater level fluctuations is stronger than that in river runoffs.

Unlike the general finding of fBm type behavior in groundwater level fluctuations (Li and Zhang, 2007; Little and Bloomfield, 2010; Rakhshandehroo and Amiri, 2012; Yu et al., 2016), Joelson et al. (2016) found persistent scaling behavior in the analysis of hourly groundwater level fluctuation time series for 14 months duration, and fit the fluctuations data with the Lévy stable distribution to account for the observed non-Gaussian heavy tailed behavior.

Multiscale Multifractal Analysis (MMA) was proposed on the basis of MF-DFA, which normally analyzes time series with crossovers only on a predefined large or small scale, to obtain the generalized Hurst surface, which simultaneously provides local fractal properties at various scale ranges (Gierałtowski et al., 2012; Wang et al., 2014). To the best of our knowledge, MMA has not yet been applied to analyze time series in hydrology or subsurface hydrology.

In this paper, DFA, MF-DFA and MMA are applied to systematically evaluate the temporal fractal scaling properties (monofractatility and multifractality) of groundwater level fluctuations in two confined aquifer wells with daily data of 70 and 80 years in Texas, USA. We also check the variation of the Hurst exponent with different lengths of data and variable time intervals. The possible explanation of the existence of multifractality is studied. Furthermore, we investigate the groundwater level fluctuation probability distribution by fitting the data with the $\alpha$-stable distribution and other distributions, such as Gaussian distribution, Gamma distribution, Lognormal distribution. Additionally, we compare the Hurst exponent from fractal analysis with that from the stability index of the fitted $\alpha$-stable distribution, since the stability index and the Hurst exponent are related under certain conditions.

## 2 Methodology

Since the pioneering work of Hurst (1951) on long memory behavior (or persistent fractal) of storage capacity of reservoirs in the Nile River, the Hurst exponent has been regarded as the best-known estimator indicating the magnitude of long-range dependence in time series, and has been widely used to study fractal scaling behavior in geophysical sciences, specifically for river flows and turbulence (Nordin et al., 1972; Szolgayova et al., 2014; Vogel et al., 1998), porosity and hydraulic conductivity in sub-surface hydrology (Molz and Boman, 1993), climate variability(Bloomfield, 1992; Franzke et al., 2015; Koutsoyiannis, 2003), and sea level fluctuations (Barbosa et al., 2006; Ercan et al., 2013).The Hurst exponent $H$ may be defined as follows:

$$\phi(ct) \overset{d}{\Rightarrow} c^H \phi(t), \forall t \geq 0, \forall c > 0 \qquad (1)$$

where $\phi$ is a given stochastic process, and c is a positive constant, and d is the finite dimension of the time series data. $0 < H < 0.5$ demonstrates anti-persistent behavior and $H{=}0.5$ corresponds to uncorrelated noise. $0.5 < H < 1$ indicates long-range dependence (i.e. persistent behavior) and $H{=}1$ is for pink noise.

Here, the Hurst exponent was adopted to quantify the scaling properties of groundwater level fluctuation time series. Many methods for the estimation of the Hurst exponent are used in the literature, and different



methods may provide significantly different estimates. Detrended fluctuation analysis is chosen here due to its superior performance compared to conventional methods in detecting evolving nonstationarities and in differentiating seasonal trends from the inherent fluctuations of time series (Yu et al., 2016).

**2.1 Detrended Fluctuation Analysis**

Detrended fluctuation analysis (DFA), also known as variance of the regression residuals, was proposed by Peng et al. (1994). The method is briefly summarized as follows:

Firstly the original time series $\{x_t\}, t = 1, 2, \cdots, n$, are converted to corresponding sums as:

$$X_t = \sum_{i=1}^{t} \left( x_i - \overline{x} \right) \tag{2}$$

Then $\{X_t\}$ is divided into $m$ ($m=n/l$) non-overlapping blocks $\{Y_j\}$ of size $l$, and a least-squares fit (or the local trend) is performed by calculating the variance for each block:

$$V = \frac{1}{l} \sum_{j=1}^{l} \left[ Y_j - Y_l(j) \right]^2 \tag{3}$$

where $Y_l(j)$ is the local fitted polynomial trend of first-order, second-order or any other higher order. Finally, the root-mean-square over all blocks is calculated, yielding the "fluctuation":

$$F(l) = \sqrt{\frac{1}{m} \sum_{i=1}^{m} V} \tag{4}$$

Fitting a linear line of $\log(F(l))$ against $\log(l)$ would indicate the presence of power-law scaling as:

$$F(l) \sqcup l^a \tag{5}$$

For fractional Gaussian noise (FGN), $\alpha = H$, where $H$ is the Hurst exponent. For non-stationary processes (e.g. fractional Brownian motion) $\alpha = H + 1$ (Heneghan and McDarby, 2000). In this study, the local trend

is fitted by a linear line. The DFA method does not assume stationarity in advance. Moreover, it is less sensitive to trends within the data than other approaches, such as the R/S, since a linear regression fit is applied locally in each block.

**2.2 Multiscale Multifractal Analysis**

Multiscale Multifractal Analysis (MMA) is a generalization of Multifractal Detrended Fluctuation Analysis (MF-DFA) which is developed from DFA(Gierałtowski et al., 2012). In contrast to MF-DFA, which requires presumption of scaling ranges, MMA is capable of concurrently characterizing different fractal properties (monofractality or multifractality) of time series over a wide range (both small and large) of temporal scales. MMA can be specified as follows:

Based on DFA, the $q$th order fluctuation is calculated as (Kantelhardt et al., 2002):



$$F_q(l) = \left\{ \frac{1}{2m} \sum_{i=1}^{2m} \left[ \frac{1}{l} \sum_{j=1}^{i} \left( Y_j - Y_i(j) \right)^2 \right]^{q/2} \right\}^{1/q} \qquad (6)$$

If long-range power-law correlation exists in the time series, then $F_q(l)$ for large values of $l$ yields (Kantelhardt et al., 2002),

$$F_q(l) \sim l^{h(q)} \qquad (7)$$

where $h(q)$ is the generalized Hurst exponent and values of $h(q)$ can be interpreted as follows: $h \in (0,0.5)$ indicates anti-persistent behavior of the time series, $h=0.5$ denotes uncorrelated noise, $h \in (0.5,1)$ indicates persistent behavior of the time series, $h=1.5$ corresponds to Brownian motion, and $h \geq 2$ indicates black noise. $h(q)$ yields the classical Hurst exponent $H$ when $q = 2$ for stationary time series and $H = h(2) - 1$ for non-stationary time series. $h(q)$ is independent of $q$ for monofractal data and strongly depends on $q$ for time series showing persistent multifractal behavior.

The strength of multifractality may be further measured by the Hölder spectrum or singularity spectrum (Feder, 1988). The Hölder exponent $\alpha_q$ and the Hölder spectrum (singularity spectrum) $f(\alpha_q)$ can be computed as follows (Kantelhardt et al., 2002):

$$\tau_q = q h(q), \alpha_q = \frac{d\tau_q}{dq} \text{ and } f(\alpha_q) = q\alpha_q - \tau_q \qquad (8)$$

where $\tau_q$ is the classical multifractal scaling exponent. The strength of multifractality in a time series can be estimated by the width of $f(\alpha_q)$, which can be illustrated by the range of $\alpha_q$ as $\Delta\alpha_q = \alpha_{max} - \alpha_{min}$ (Koscielny-Bunde et al., 2006).

The above estimators show the formulation of MF-DFA. After the calculation of all $F_q(l)$ by MF-DFA, a moving fitting time window, which completely sweeps through the range of scale $l$ along $F_q(l)$, is used to study quasicontinuous changes between $h(q)$ dependence and the range of scale $l$. The fitting procedure is as follows:

$$h_{f_i}(q,l) = \frac{\log\left[ \Delta F_q(l)_{f_i} \right]}{\log\left( \Delta l_{f_i} \right)} \qquad (9)$$

where $f_i$ is a fitting window ($i = 1, 2, \ldots, n$) and $h_{f_i}$ is the local scaling exponent in $f_i$. For a fixed $q$, the spectrum of scaling exponents over the whole range of scale $l$ is obtained by $h(q,l) = \left\{ h_{f_1}, h_{f_2}, \ldots, h_{f_n} \right\}$.

After plotting the results of $h(q,l)$ for all the $q$, the Hurst surface $h(q,l)$, which simultaneously provides the generalized Hurst exponent for multiple scales and $q$, is obtained (Wang et al., 2014).

The capability of MMA, which is inherited from DFA and MF-DFA, is that it can effectively detect observational noise and nonstationarities in time series. Similar to MF-DFA, the results of $h(q,l)$ in MMA





characterize large fluctuations in the fragments of data for $q > 0$, while the results of $h(q, l)$ correspond to small fluctuations for $q < 0$.

### 2.3 Alpha-Stable Distributions

The $\alpha$-stable distributions, introduced by Lévy (1925), represent a class of stable laws determined by four parameters: the stability index $\alpha$, the skewness parameter $\beta$, the scale parameter $\gamma$ and the location parameter $\delta$. Therefore, the $\alpha$-stable distribution of a random variable X is usually denoted by $X \sim S_\alpha(\beta, \gamma, \delta)$. No closed forms exist for the $\alpha$-stable distributions, except for the following three distributions: Gaussian, Cauchy and Lévy. The $\alpha$-stable distribution of a random variable, $X \sim S_\alpha(\beta, \gamma, \delta)$, can be described by the following

characteristic function (Samoradnitsky and Taqqu, 1994):

$$\phi_x(t) = \begin{cases} \exp\left\{-\gamma^\alpha |t|^\alpha \left[1 - i\beta \, sign(t)\tan\left(\frac{\pi\alpha}{2}\right)\right] + i\delta t\right\}, \alpha \neq 1 \\ \exp\left\{-\gamma^\alpha |t|^\alpha \left[1 + i\beta \, sign(t)\left(\frac{2}{\pi}\right)\log|t|\right] + i\delta t\right\}, \alpha = 1 \end{cases} \tag{10}$$

where

$$sign(t) = \begin{cases} 1, \, if \, t > 0 \\ 0, if \, t = 0 \\ -1, if \, t < 0 \end{cases}$$

The stability index $\alpha$ is also known as the characteristic exponent and is in the interval of $\alpha \in (0,2]$. The distribution becomes normal distribution when $\alpha = 2$. The skewness parameter satisfies $-1 \leq \beta \leq 1$. The

location parameter $\delta$ indicates the shift of the peak of the distribution and it is undefined unless $\alpha > 1$. The distribution is symmetric around $\delta$ if $\beta = 0$. The scale parameter $\gamma$ measures the dispersion of the distribution and is always positive ($\gamma > 0$).

Stable distributions are heavy-tailed, and tails of these distributions demonstrate asymptotical power law behavior with $0 < \alpha < 2$ and $-1 < \beta < 1$. One important property of the $\alpha$-stable distribution is that there

is a possible link between the stable distribution and self-affine behavior, according to the generalized central limit theorem(Gnedenko and Kolmogorov, 1956). To be more specific, approximation of the tail of the stable distribution $X \sim S_\alpha(\beta, \gamma, \delta)$ may be shown (Samoradnitsky and Taqqu, 1994):

$$\begin{cases} \lim_{x \to \infty} P(X > x) = c_\alpha (1 + \beta)\gamma^\alpha x^{-\alpha} \\ \lim_{x \to \infty} P(X < -x) = c_\alpha (1 - \beta)\gamma^\alpha x^{-\alpha} \end{cases} \tag{11}$$

where $c_\alpha = \frac{1}{\pi} \sin\left(\frac{\pi\alpha}{2}\right)\Gamma(\alpha)$ . This behavior indicates that $\alpha$-stable distributions can be well accommodated to model self-similar processes. The distribution with $1 < \alpha < 2$ is of significant interest to

researchers as the mean of the distribution can be defined and the variance is infinite. The non-integer $\alpha$ in



this range, which is capable of characterizing processes with infinite variance, is related to Hurst exponent $H$, presenting long-range dependence and statistical self-similarity properties, as follows (Taqqu et al., 1997):

$$\alpha = 3 - 2H \tag{12}$$

Since the Lévy $\alpha$-stable distribution is $1/\alpha$-self-similar, the following equation is also used to describe the relationship between the stability index and the Hurst exponent (Peters, 1994):

$$\alpha = \frac{1}{H} \tag{13}$$

## 3 Data Analysis

Two confined aquifer wells with long groundwater level records (70 and 80 years long) were chosen in this study to perform fractal scaling analysis. Groundwater level time series data of the two wells were obtained from the Water Data for Texas website (http://waterdatafortexas.org/groundwater/). Geophysical properties and basic statistics of the groundwater levels of the two wells are listed in Table 1. Based on the data availability, the study period was chosen from January 1, 1945 to December 31, 2014 for Well1, and from January 1, 1935 to December 31, 2014 for Well2. The missing groundwater level data of the two wells were obtained by linear interpolation. The total daily records used in this study are 25,567 and 29,220, for Well 1 and Well 2 respectively (Fig.1).

The autocorrelation function (ACF) in Fig.2 shows very slow decay in both datasets, and the dataset of Well1 decays more slowly than that of Well2. In fact, it takes several years (more than 1000 days) for Well1 to become decorrelated while it takes a couple of years (more than 500 days) for Well2. Moreover, the ACF plots greatly vary in different 20-year intervals of the two datasets (bottom left and right figures of Fig. 2), which may imply that the long-range dependence characteristics of the two wells would vary through time. The power spectra of Well1 (1945-2014) and Well2 (1935-2014) groundwater levels are presented in Fig. 3. The power-law exponents are estimated as 2.44 and 2.08 for Well1 and Well2 groundwater levels, respectively, indicating the existence of fractals in both datasets. Hurst exponents can be deduced from the power-law exponents (Heneghan and McDarby, 2000) as 0.72 and 0.54 for Well1 and Well2 groundwater levels, respectively. Furthermore, Kwiatkowski-Phillips-Schmidt-Shin (KPSS) test (Kwiatkowski et al., 1992) is conducted to test the stationarity of data. The null hypothesis for KPSS test is that a time series is stationary and the alternative is that data are non-stationary. The estimates of KPSS statistic are 5.6357 and 1.8012 for Well1 and Well2 groundwater levels respectively, and both reject the null hypothesis at 1% significance level, which suggests that the two time series are non-stationary. These results provide reference for the quantification of the Hurst exponent later by DFA.

### 3.1 Monofractal Analysis

The Hurst exponents of groundwater level fluctuation data, quantified by DFA approach over different time intervals are investigated here. The evolution of Hurst exponent $H$ through time is shown in Fig. 4, where the





data were chosen in the original order, moving year by year forward from 1945 to 2014 for Well1 (i.e.. 1945-1949, 1945-1950,…, 1945-2014) and from 1935 to 2014 for Well2 (i.e.. 1935-1939, 1935-1940,…, 1935-2014). Figures 4a and 4b clearly show that Hurst exponent varies through time for both Well1 and Well2 groundwater levels. Boxplots in Fig. 4c demonstrate that the mean and variance of the Hurst exponent

through time differ noticeably for both Well1 and Well2 groundwater levels. $H$ is 0.73 and 0.51 for Well1 and Well2, respectively, when all the available data are used, which suggests that both datasets indicate long memory. These estimates of Hurst exponents are also consistent with the ones that are deduced from the power-law exponents in Fig. 3. In general, groundwater level fluctuations of Well1 show persistent fractal behavior ($H > 0.5$, more specifically $H > 0.7$,) for all investigated time periods, and those of Well2 vary between persistent and anti-persistent, even showing uncorrelated behavior at certain times.

The Hurst exponent $H$ for Well1 groundwater levels varies between 0.8 and 0.85 for up to 8 years of daily data for end years 1948-1952 (Fig. 4a), and then stabilizes within values of 0.71 and 0.78 for 9 years and longer time durations (for end years greater than 1952 in Fig. 4a). On the other hand, $H$ for Well2 groundwater levels varies between 0.53 and 0.6 for up to 7 years of daily data for end years 1939-1941 (Fig. 4b), and then

stabilizes within 0.46 and 0.53 for longer time durations than 8 years (for end years greater than 1941 in Fig. 4b). As such, fractal behavior of groundwater levels, obtained from short duration data (in this study, less than 8 years for Well1 and 7 years for Well2), may not exhibit the stable long-term fractal behavior. These results further imply that the length of time series and the time period it covers jointly affect the value of $H$. The Hurst exponents here demonstrate the ability of DFA in distinguishing the seemingly long-range

correlations caused by external effects (such as seasonal trend) from its intrinsic fluctuations (Yu et al., 2016), since the ACF plots show very slow decay in both wells (Fig.2).

Figure 5 presents the Hurst exponents of groundwater level data estimated with different moving time windows (5-year, 10-year and 20-year). Daily data were used in different time windows: 5-yr moving window (i.e., 1945-1949, 1946-1950, …, 2010-2014), 10-yr moving window (i.e.,1945-1954, 1946-1955, …, 2005-

2014), and 20-yr moving window (i.e., 1945-1964, 1946-1965, …, 1995-2014). Figures 5a and 5b show that the Hurst exponents vary greatly in different time windows (i.e.. different length of groundwater level fluctuation data), and also do not remain constant even with the same time window when the time window moves in time. Moreover, the results in Figures 5a and 5b demonstrate that the Hurst exponent tends to be stable as the time window increases, which is consistent with the results in Fig. 4.

Additionally, the correlation coefficient $r$ is used to investigate the relationship between the Hurst exponent and the variation in groundwater level fluctuations, which is quantified by the coefficient of variation, $c_v$ ($c_v = \delta/\mu$, where $\delta$ is the standard deviation of the data and $\mu$ is the corresponding average). From Fig. 5c it may be inferred that strong linear correlation exists between $c_v$ and $H$ ($r > 0.5$), and the correlation becomes stronger as time window increases from 5 to 20. Meanwhile, for Well2 groundwater levels the

correlation is weaker ($r < 0.5$), and $r$ increases first and then decreases afterwards, following the increase of time window from 5 to 20 (Fig. 5d). For Well1 groundwater levels (Fig. 5c), a larger $c_v$ normally follows a greater $H$ for 5, 10 and 20-yr time windows. However, for Well2 groundwater levels (Fig. 5d), this





relationship generally does not hold (especially for the 20-yr time window). Figures 5c and 5d suggest that the variability of groundwater level fluctuation may affect the intrinsic correlation (long memory or short memory) of the data, but it is highly site-specific. The different Hurst exponents in different wells may be due to the effect of heterogeneity of the aquifer materials (Li and Zhang 2007).

Figure 6 further investigates the variation of the Hurst exponents by the boxplots of 5, 10, 20, 30, 40, and 50-yr moving time windows. Unlike the inconsistency of the linear correlation between $c_v$ and $H$, the variation of $H$ in both Well1 and Well2 groundwater levels are consistent here. The variation of $H$ for both wells' groundwater levels, in general, decreases as the moving time window increases, which confirms the findings in Figures 5a and 5b.

**3.2 Multifractal Analysis**

The multifractal results obtained by MF-DFA in Fig.7 include log-log plots of $F_q(l)$ against time scale $l$, the generalized Hurst exponent $h(q)$, the scaling exponent spectrum $\tau_q$ and the singularity spectrum $f(\alpha_q)$ corresponding to a series of moments $q(-5 \leq q \leq 5)$. $h(q)$ is the slope of the linear regression line of the log-log plot for a given $q$ . Clearly, multifractality exists in groundwater levels of Well1 (1935-2014) and

Well2 (1945-2014), since $h(q)$ greatly varies with $q$, as demonstrated in Figures 7a and 7b, and the relationships between $\tau_q$ and $q$ in Fig. 6c are not linear for groundwater levels of Well1 and Well2. This also suggests that different exponents should be used to illustrate the fractal scaling behavior (self-affinity) of different time intervals of the data. Moreover, $h(q)$ continuously decreases as $q$ increases in both figures, implying that relatively small fluctuations occur more frequently in the time series than the large ones (Grech

and Czarnecki, 2009; Rakhshandehroo and Amiri, 2012).

The singularities of the processes in the groundwater levels of Well1 and Well2 are revealed in Fig. 7d. The width of the singularity spectrum, $\Delta\alpha_q$, is used to measure the level of multifractality. The width of the singularity spectrum $\Delta\alpha_q$ tends to be zero for monofractal structures, and would increase as the level of multifractality of the signal increases. $\Delta\alpha_q$ was found to be 4.05 for the groundwater levels of Well1 and 1.07

for the groundwater levels of Well2. These results indicate a high level of multifractality in both time series, and Well1 groundwater levels have a stronger multifractality, which further suggests that the multifractal behavior is quite site-specific.

Two types of rationale are used to account for multifractality in time series (Kantelhardt et al., 2002). The first type is that a broad probability density function of time series data, which cannot be represented by a

regular distribution with finite moments, causes multifractality. The second type is that multifractality is caused by long-range correlations of small and large fluctuations (Kantelhardt et al., 2002; Rakhshandehroo and Amiri, 2012).To distinguish these two types of multifractality, the corresponding randomly shuffled dataset is analyzed. The multifractality will vanish if it is totally due to the second type and will remain otherwise. If the multifractality is due to both types, the shuffled data will present weaker multifractality than

the original data (Kantelhardt et al., 2002).





Therefore, a shuffling procedure was conducted to investigate the types of the multifractality for Well1 and Well2 groundwater levels. The corresponding multifractality results are shown in Fig. 8. This figure clearly shows that multifractality still exists in the shuffled groundwater level data of Well1, since dependency between $h(q)$ and $q$ remains (Fig. 8a). The relationship between $\tau_q$ and q is not linear (Fig. 8c), which

further verifies the existence of multifractality in shuffled Well1 data $\Delta\alpha_q$ was 0.18 (Fig. 8d), which indicates a much weaker multifractality compared with $\Delta\alpha_q = 4.05$ for the original data. The results for shuffled Well2 groundwater level data, on the contrary, show that shuffling almost completely destroyed its intrinsic fractal correlations, since $h(q)$ is independent of q (Fig. 8b), $\tau_q$ is linear with q (Fig. 8c), and the singularity spectrum almost converges to a single point with $\Delta\alpha_q = 0.02$ (Fig. 8d), which may indicate an approximate

monofractal structure in Well2 groundwater levels.

Results in Fig. 8 reveal that different types of multifractality exist in Well1 and Well2 groundwater level time series. For Well1, the multifractality is clearly due to the combined effect of a broad probability density function and temporal correlations in diverse magnitudes of fluctuations. Meanwhile, the multifractality is almost purely caused by long-range temporal correlations in small and large fluctuations for Well2

groundwater levels.

Since the Hurst exponent varies for different time intervals of the groundwater level time series of Well 1 and Well 2 (Figures 4, 5, and 6), and the finding that the small and large flutuations of temporal correlations contribute to multifractality of both datasets (Fig. 8), the Multiscale Multifractal Analysis (MMA) is adopted to investigate the fractal behavior at different temporal scale ranges in detail, as demonstrated in Fig. 9. It is

noted that the generalized Hurst surfaces for the original datasets of both Well1 and Well2 groundwater levels (top figures of Fig. 9) are far from flat (hill-like shape), which clearly suggests different fractal scaling exponents are needed to represent fractal behavior at multiple temporal scales for both datasets. In addition, the generalized Hurst exponents at $q$=2 are between 1.5 and 2 for Well1 groundwater levels, indicating persistent behavior, and are mostly within the range between 1 and 1.55 for Well2 groundwater levels,

indicating persistent and anti-persistent fractal behavior (sometimes even uncorrelated). Moreover, the Hurst surfaces for the shuffled time series of Well1 and Well2 ( bottom figures of Fig. 9) show that the surfaces become much flatter than those generated by the original datasets (small fluctuations of the Hurst surfaces still exist after shuffling),which suggests that the shuffling substantially destroys the intrinsic correlations, as consistent with the MF-DFA results.

**3.3 Relationship between the stability index and the Hurst exponent**

Multifractal analysis suggests that the multifractality is partially due to a broad probability density distribution that may have infinite moments. However, fBm (fractional Brownian motion), which has been identified as an appropriate model to characterize groundwater level fluctuation (Li and Zhang, 2007; Little and Bloomfield, 2010; Rakhshandehroo and Amiri, 2012; Yu et al., 2016) is Gaussian with finite moments.

Therefore, fBm may be inappropriate for the description of physical processes with infinite moments, such as the groundwater level fluctuations in this study. Histograms and Normal probability plots for Well1 and



Well2 groundwater levels in six selected durations of varying length apparently indicate that the Gaussian distribution may not be suitable to represent the groundwater level processes of both wells, especially for Well1 (Figures 10 and 11). Well1 groundwater levels clearly show heavy tail, and Well2 groundwater levels demonstrate right-skewed behavior. As such, the Lévy alpha stable distribution, which is non-Gaussian with

heavy tail and has infinite variance, was adopted to fit the groundwater datasets. Moreover, to obtain a relatively comprehensive picture of the underlying probability distribution, Gaussian distribution, Gamma distribution and Lognormal distribution were also used to fit the datasets (the Statistics and Machine Learning Toolbox in Matlab are used for this purpose). The fitting procedure is conducted continuously with the data starting from 1945 for Well1 and from 1935 for Well2, and moves forward year-by-year with the same end

year, 2014, for all the fitting durations (at least 15 years of daily data are used to ensure a good characterization of the data distribution). Results for the six selected durations are presented in Figures 12 and 13 for Well1 and Well2 groundwater levels, respectively.

Figure12 shows that in general, the Lévy stable distribution fits the groundwater level fluctuation time series of Well1 over different durations very well. Meanwhile, the other distributions, i.e. Normal, Gamma and

Lognormal, cannot satisfactorily capture the behavior of the groundwater levels of Well1. This verifies the finding that the irregular distribution of Well1 groundwater levels contributes to the multifractality. For Well2 groundwater levels, Gaussian distribution adequately fits the data, except at the peak values (Fig. 13). Furthermore, the fitted stable, Gamma and Lognormal distributions converge to the Gaussian distribution. This may imply that fBm may partially represent the behavior of Well2 groundwater levels, which has the

Hurst exponent fluctuating between 0.48 and 0.52 (Fig.14b).

Furthermore, the stability index $\alpha$ of the stable distribution is related to the Hurst exponent $H$, given by a relationship between $\alpha$ and $H$. Two formulae (Eq.12 and Eq.13) are used to estimate $H$. The estimated $H$ is then compared with that estimated by DFA (Fig. 14). With respect to the difference between the Hurst exponent estimated by DFA (for both Well1 and Well2 groundwater levels) and that deducted from either

$H = \frac{1}{\alpha}$ or $H = \frac{3-\alpha}{2}$ or, the relative difference is less than 10% (even less than 1% in some time intervals) for most of the comparisons. For Well1 groundwater levels, the Hurst exponent by $H = \frac{1}{\alpha}$ generally matches better with $H$ estimated by DFA than that by $H = \frac{3-\alpha}{2}$ (Fig. 14a), although for some durations, such as from 1950-2014 to 1955-2014, the latter one performs better than the first. Fig. 14c further shows that the stability index $\alpha$ is strongly correlated with the coefficient of variation $c_v$ of the groundwater level fluctuation data

from Well1, since the correlation coefficient is as high as -0.84. It suggests that a larger $c_v$ of Well1 groundwater levels would probably imply a smaller $\alpha$. A stability index $\alpha = 2$ for all the stable distributions for the groundwater levels data from Well2 ($\alpha = 2$ corresponds to Gaussian distribution) is found. This may be due to the fact that the Lévy stable distribution here is restricted in the range $1 < \alpha \leq 2$, which corresponds to $0.5 \leq H < 1$. However, Well2 groundwater levels do not have long memory in some time

intervals. The relationship between $H$ and $\alpha$ would completely fail when $H<0.5$. However, the resulting stability index $\alpha = 2$ for Well2 groundwater levels is acceptable, considering the difference between $H$



estimated by DFA and *H* estimated by the stability index is less than 5% for all the time periods. This result is also consistent with Fig.11 where Gaussian distribution is capable of capturing the main groundwater level fluctuation patterns of Well2.

The results indicate that fBm, which has Gaussian characteristics, may be a reasonable model for representing groundwater level fluctuations under certain conditions, such as in the case of the dataset of Well2, which has the Hurst exponent fluctuating closely around 0.5. However, fBm may be an insufficient model for capturing the behavior of groundwater fluctuations in other cases, for example in the case of the groundwater levels of Well1, where a non-Gaussian distribution, such as a heavy-tailed stable distribution (Lévy motion), is needed instead. In the presence of long-memory, fractional Lévy motion may be more appropriate to model and forecast the groundwater dynamics.

It is important to note that the results obtained so far are limited to the analysis of temporal correlations of the groundwater level fluctuations at certain locations. The properties of the groundwater levels at two wells, such as their fractal behavior and underlying distributions, are highly different from each other, which confirms that the results are site-specific. Groundwater dynamics in aquifers result from multiple complex dynamic processes, such as the hydrologic processes (precipitation and river runoff, etc), hydraulic properties of soil and aquifers, and anthropogenic perturbations (such as construction of reservoirs and pumping of water). These processes and properties vary at different spatio-temporal scales, which directly or indirectly affect groundwater systems. The results herein may be attributed to the time-space heterogeneity of aquifer characteristics, but detailed research, such as the employment of time-space analysis, needs to be conducted to justify this and to account for the effect of heterogeneity on fractal behavior at different temporal scales. Non-Gaussian fractal property of the groundwater system in Well1 that demonstrates long memory, provides further insight for the resulting transport processes in the porous medium, which may also present non-Gausssian features with memory, similar to the non-Gaussian behavior that is found in the precipitation time series in other studies (Joelson et al., 2016; Lovejoy and Mandelbrot, 1985). Unlike Well1 groundwater levels, the origin of multifractality for Well2 groundwater levels is difficult to explain, due to the very weak multifractality after the shuffling. An intuitive explanation may be that it is due to noise. However, the fractal structure is not affected by dynamical noise (Serletis, 2008). Additionally, Gaussian distribution may partially represent the dataset of Well2 groundwater levels, but it fails to capture the peak of the skewed distribution of Well2 groundwater levels, which may imply that an irregular distribution that also holds certain Gaussian characteristics may be needed to fully characterize the groundwater dynamics of Well2.

**4 Conclusions**

In this study, fractal scaling properties of groundwater level fluctuations of two confined aquifer wells, with 70 and 80 years of daily data, were analyzed. Detrended fluctuation analysis (DFA) was utilized to quantify the Hurst exponent and monofractality. The DFA results indicated that fractals exist in groundwater level time series of both wells, and it was shown that the Hurst exponent is closely dependent on the length and specific time period of the time series. Persistent scaling pattern was found for all investigated time periods



of Well1 groundwater levels (Hurst exponent, $H>0.5$), and the scaling pattern varied between anti-persistent and persistent regimes for Well2 groundwater levels. The Hurst exponent $H$ for Well1 groundwater levels fluctuated between 0.8 and 0.85 for up to 8 years of daily data for end years 1948-1952 (Fig. 4a), and then stabilized within the range of 0.71 - 0.78 for 9 years and longer time durations (for end years greater than

1952 in Fig. 4a). On the other hand, $H$ for Well2 groundwater levels fluctuated between 0.53 and 0.6 for up to 7 years of daily data for end years 1939-1941 (Fig. 4b), and then stabilized within the range 0.46 - 0.53 for durations longer than 8 years (for end years greater than 1941 in Fig. 4b).

Multifractal detrended fluctuation analysis (MF-DFA) and Multiscale Multifractal Analysis (MMA) were adopted to examine the multifractality and multifractal behavior at different temporal scales for confined

groundwater levels. Although the MF-DFA results showed that relatively high level of multifractality exists for both wells' groundwater levels, a stronger multifractality was observed for the dataset of Well1 compared to that of Well2. The observed multifractality was postulated to originate from the combined effect of the underlying irregular probability distributions and different magnitudes of fluctuations on multiple long-range temporal correlations for Well1 groundwater levels, and mostly long-range temporal correlations in small

and large fluctuations for Well2 groundwater levels. Moreover, the MMA results confirmed the existence of multifractality and diverse correlations of groundwater levels over different time scales. Furthermore, the underlying probability distribution of groundwater level fluctuations for Well1 represented mainly long memory characteristics, which were fitted reasonably well by the Lévy stable distributions for various time periods. On the other hand, those of Well2 represented mainly Gaussian characteristics, which sometimes

failed to capture the peaks of the skewed probability distributions of Well2 groundwater levels. Time series analysis of groundwater level fluctuations of the two wells demonstrated that the observed fractal behavior is site-specific, and there is a need for generalized governing equations of groundwater flow processes, which can model both the long-memory behavior as well as the Brownian finite-memory behavior.

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



Table 1. Statistics and geophysical properties of studied wells in Texas, USA

| Well | ID | Location (lat, long) | Well Depth Below Land Surface (m) | Aquifer Type | Record Period | Mean (m) | STD(m) |
|------|-----|---------------------|-----------------------------------|--------------|---------------|----------|--------|
| Well1 | 6950302 | (29.208888° N, 99.784444° W) | 87.48 | Confined | 1940-10-24 to present | 11.57 | 4.75 |
| Well2 | 6837203 | (29.479166° N, 98.432499° W) | 266.40 | Confined | 1932-11-12 to present | 20.10 | 5.00 |

Note: Mean represents the mean groundwater level (hydraulic head) depth below land surface.





**Figure 1. Groundwater level time series data of (a) Well1and (b) Well2**

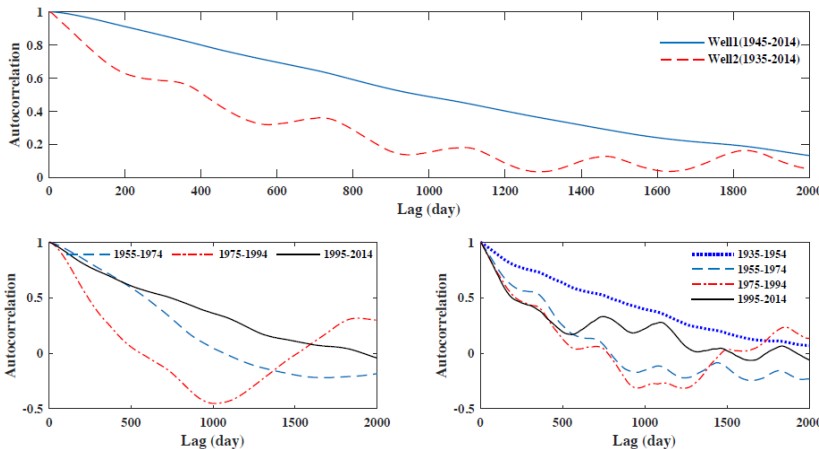

**Figure 2. Autocorrelation function (ACF) of the whole groundwater level datasets (Top); ACF at every 20 years interval for Well 1 (bottom left) and Well 2 (bottom right).**





**Figure 3. Power spectra of (a) Well1 and (b) Well2 groundwater levels.**

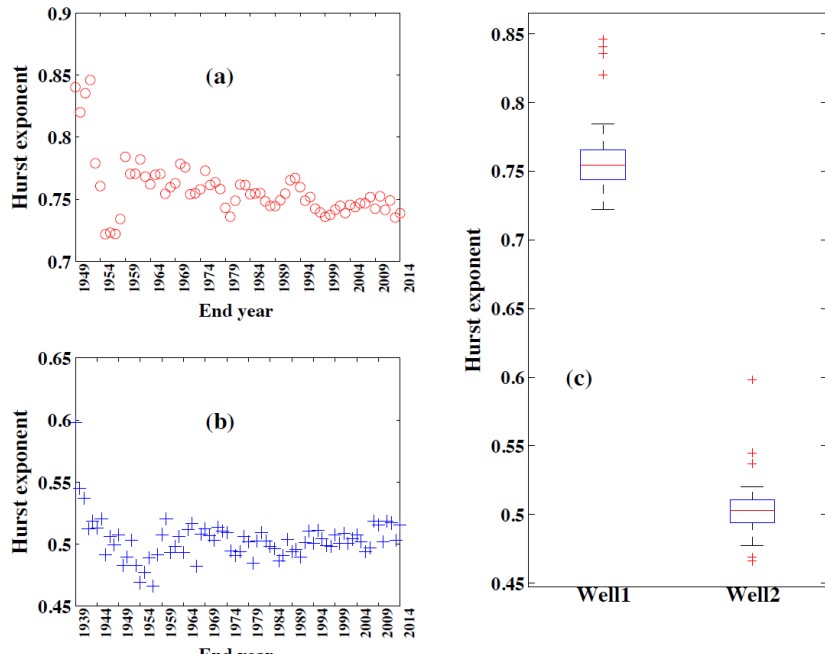

**Figure 4. (a)The evolution of the Hurst exponent through time for Well1 groundwater levels, which starts from 1945; (b)The evolution of the Hurst exponent through time for Well2 groundwater levels, which starts from 1935; (c) Boxplots of the Hurst exponents in (a) and (b).**



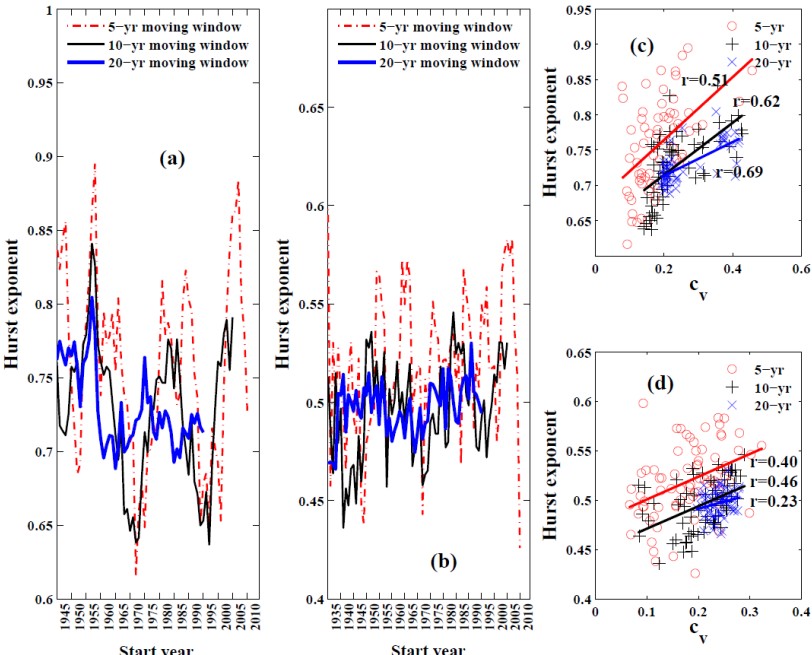

**Figure 5.** **(a) The Hurst exponent of Well 1 groundwater levels, estimated by DFA within different time windows. (b) The Hurst exponent of Well 2 groundwater levels, estimated by DFA within different time windows. (c) The relationship between the coefficient of variation $c_v$ and the Hurst exponent obtained in (a). (d) The relationship between the coefficient of variation $c_v$ and the Hurst exponent obtained in (b).**



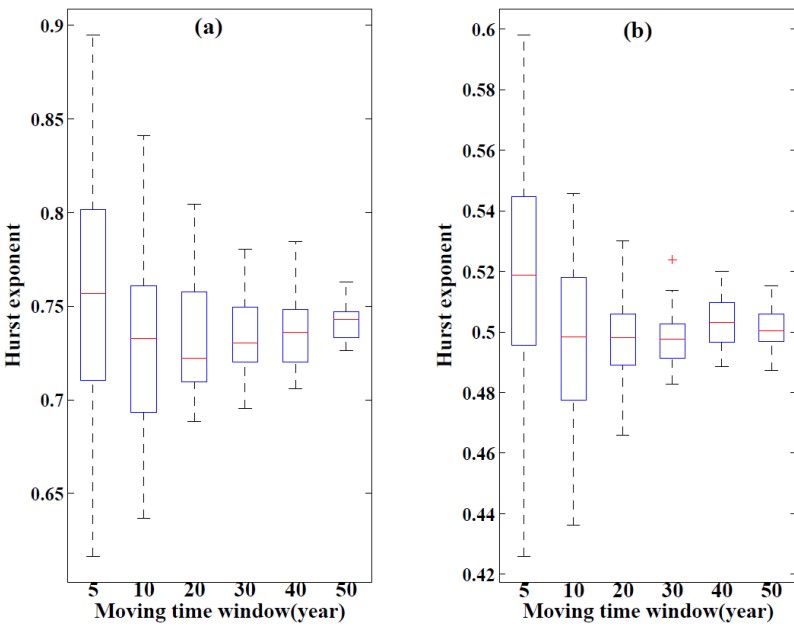

**Figure 6. Boxplots of the Hurst exponents under different moving time windows for (a) Well1 groundwater levels and (b) Well2 groundwater levels.**

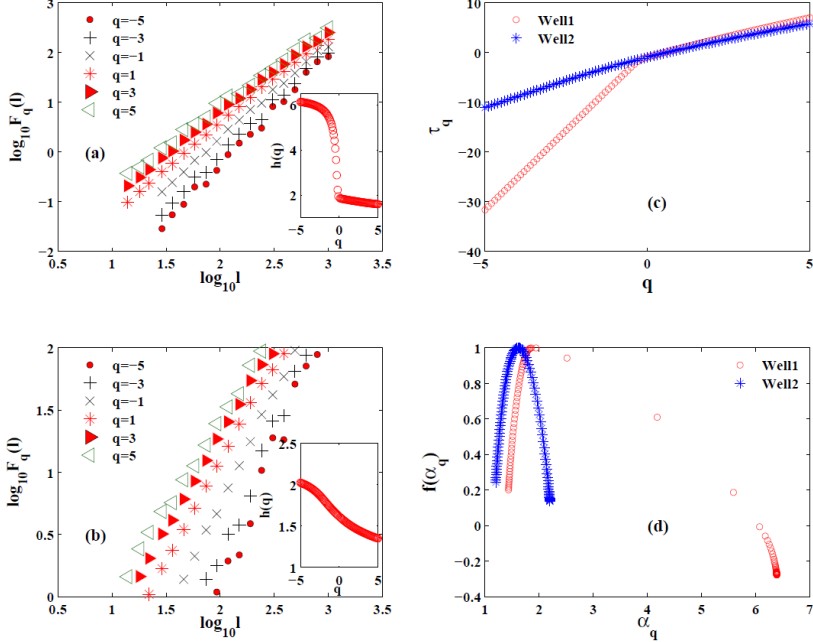

**Figure 7.** $F_q$ **as a function of time scale l and the generalized Hurst exponent h as a function of q for the groundwater levels of (a) Well1 (b) Well2; (c) the scaling exponent spectrum** $\tau_q$ **vs the moments for the groundwater levels of Well1 and Well2; (d) the singularity spectrum for the groundwater levels of Well1 and Well 2.**





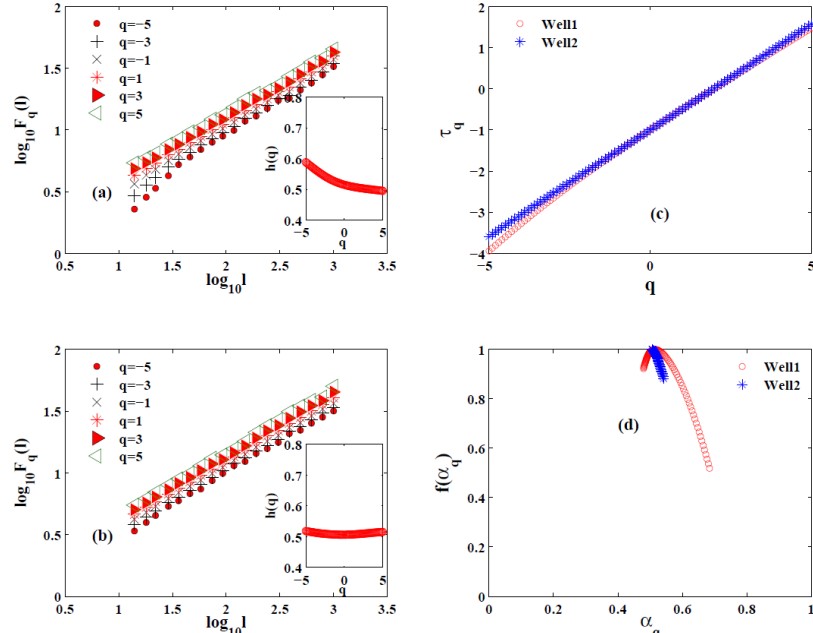

**Figure 8.** $F_q$ **as a function of time scale l, and the generalized Hurst exponent h as a function of q after shuffling for (a) Well1 groundwater levels, (b) Well2 groundwater levels, (c) the scaling exponent spectrum $\tau_q$ vs the moments for Well1 and Well2 groundwater levels after shuffling, and (d) the singularity spectrum for Well1 and Well 2 groundwater levels after shuffling.**



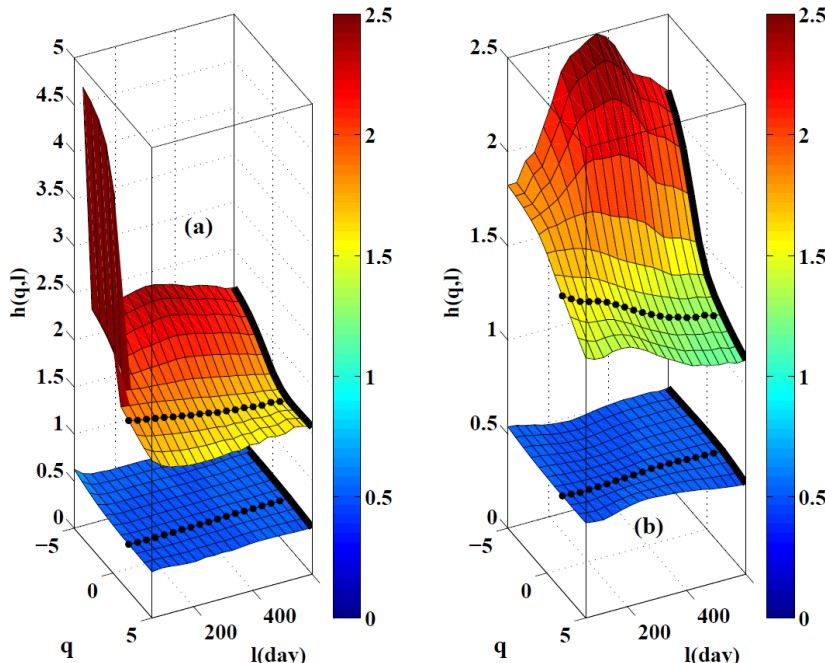

**Figure 9. Multiscale Multifractal Analysis for (a) Well1 groundwater levels and (b) Well2 groundwater levels . The top figure is for the original data and the bottom is for the shuffled data. The thick black line indicates MF-DFA results for a given temporal scale, and the solid dots show the generalized Hurst exponents at q=2 by DFA over different scales.**





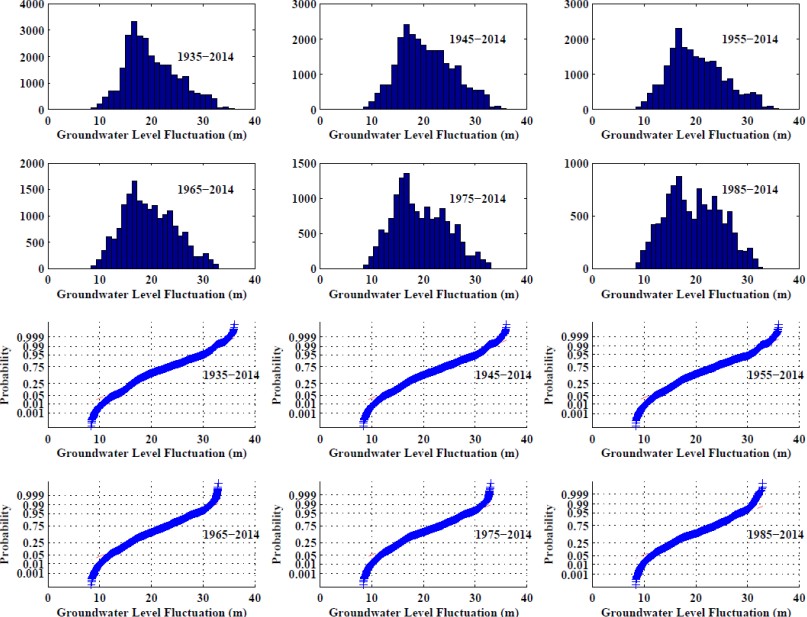

**Figure 10. Histograms and Normal probability plots for various time intervals of groundwater levels of Well1**

**Figure 11. Histograms and Normal probability plots for various time intervals of groundwater levels of Well2**





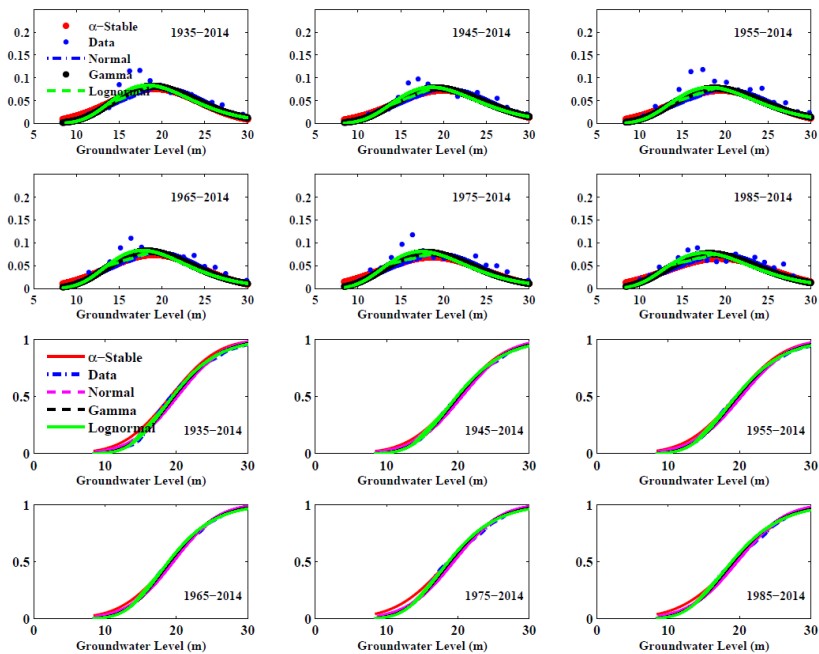

**Figure 12. Probability density function and cumulative distribution function (first two lines and last two lines respectively) of groundwater level fluctuation time series data of Well1.**

**Figure 13. Probability density function and cumulative distribution function (first two lines and last two lines respectively) of groundwater level fluctuation time series data of Well2.**



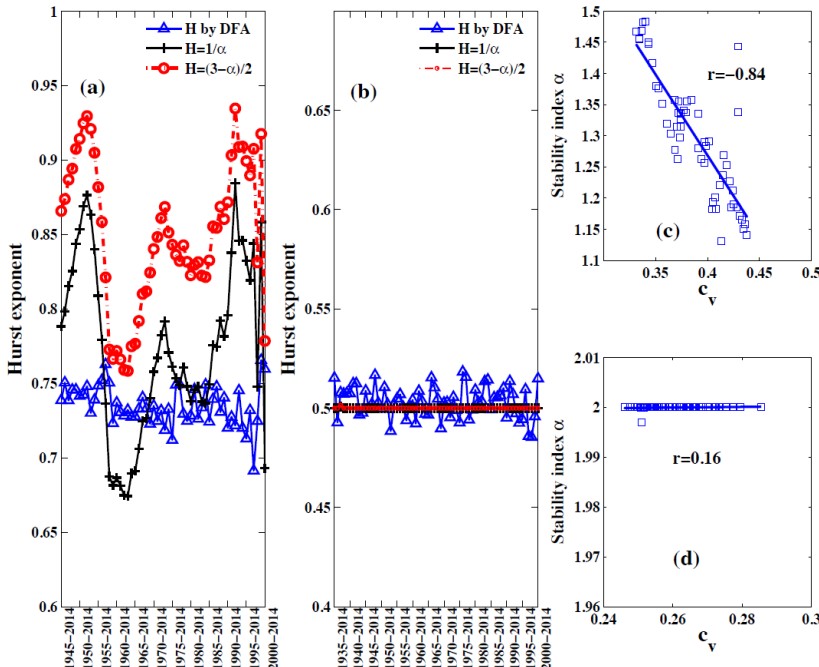

**Figure 14. (a) Well1 groundwater levels: Comparison between the Hurst exponent estimated by DFA and that by stability index. (b) Well2 groundwater levels: Comparison between the Hurst exponent estimated by DFA and that by stability index. (c) Well1 groundwater levels: The coefficient of variation $c_v$ versus the stability index obtained in (a). (d) Well 2 groundwater levels: The coefficient of variation $c_v$ versus the stability index obtained in (b).**

