# Peer review of "Fractal Scaling Analysis of Groundwater Dynamics in Confined Aquifers"

_Earth System Dynamics, 2017_

## Referee Comment (RC1) · Anonymous Referee #1 · 29 Apr 2017

This manuscript analyzed fractal scaling of groundwater dynamics in confined aquifers and presented important stochastic characteristics of the groundwater. This kind of stochastic analysis is very important to understand the hydrological processes of the groundwater and to improver generalized governing equations of groundwater flow processes. I strongly recommend the manuscript be published in the Journal to share the stochastic properties of the groundwater with the other readers for better understanding of groundwater flow processes in Hydrology. However, minor questions written in below should be considered to revise the manuscript before publication.

1. Well1 and Well2

Why did the authors select these two wells? Please explain how you selected these two wells. Also, please explain the relationship and geophysical and hydrological characteristics of these two wells more clearly. For example, are these two wells located in the same river basin? What is the distance from Well1 to Well2? As such, more detailed explanation should be specified for these two wells. Table 1 presents some information but it is not enough for the readers.

2. The number of wells selected in this study

Why did the authors focus only two wells? I believe there are many wells those provide long period of ground water data. As the authors mentioned, the results presented in the manuscript are site-specific. In that case, it would be much better to explain why only two wells were focused and analyzed in the manuscript.

---

## Referee Comment (RC2) · Anonymous Referee #2 · 2 May 2017

This study conducted fractal scaling analysis of groundwater level fluctuations in confined aquifer wells by means of Detrended fluctuation analysis (DFA), Multifractal detrended fluctuation analysis (MF-DFA), and Multiscale Multifractal Analysis (MMA), and by investigating the relationship between the stability index and the Hurst exponent. In my opinion, this study provides important knowledge on groundwater level fluctuations with sufficient novelty, and this paper was mostly written well. However, some parts, especially Introduction, need to be improved before publication.

1. P.2 L.8-18: The relationship between this paragraph and this study is not clear. Please add some more explanations or descriptions.

2. P.2 L.25: (MF-DFA(Kantelhardt et al., 2002) -> (MF-DFA, Kantelhardt et al., 2002)

3. P.3 L.20-21: Please add a reference for the sentence: "stability index and the Hurst

exponent are related under certain conditions."

4. P3. L.13-21: I strongly recommend the authors to explicitly explain the differences between existing related studies and this study in order to emphasize the novelty of this study. In addition, please briefly describe the purpose of each analysis here, and explain why the authors used long-term groundwater level data in this study.

5. Equation (1): Please define t.

6. P.4 L.1-3: I recommend the authors to explain in this section why it is important to detect evolving nonstationarities in this study.

7. 3 Data Analysis: How were the two wells selected? Why were only two wells used in this study?

8. Figures 10 and 11: What are the red dot lines in the normal probability plots?

9. Conclusions: Please add the summary of the investigation on the relationship between the stability index and the Hurst exponent.
* * *

---

## Author Comment (AC1) · 24 Jul 2017

**Response to Comments of Anonymous Referee #1**

The authors thank the anonymous referee #1 for the valuable and insightful comments. Responses to the issues raised by the Reviewer #1 are provided below in red color:

This manuscript analyzed fractal scaling of groundwater dynamics in confined aquifers and presented important stochastic characteristics of the groundwater. This kind of stochastic analysis is very important to understand the hydrological processes of the groundwater and to improver generalized governing equations of groundwater flow processes. I strongly recommend the manuscript be published in the Journal to share the stochastic properties of the groundwater with the other readers for better understanding of groundwater flow processes in Hydrology. However, minor questions written in below should be considered to revise the manuscript before publication.

The authors thank the positive comments of the Reviewer #1.

1. Well1 and Well2
Why did the authors select these two wells? Please explain how you selected these two wells. Also, please explain the relationship and geophysical and hydrological characteristics of these two wells more clearly. For example, are these two wells located in the same river basin? What is the distance from Well1 to Well2? As such, more detailed explanation should be specified for these two wells. Table 1 presents some information but it is not enough for the readers.
*Authors' response*:
In this study, the authors tried to find some long-term and complete groundwater level observations from the groundwater datasets that can be accessed through the webpage of Water Data for Texas. The candidate wells are expected to have more than 50 years of continuous daily records with less than 5% of missing data. This is the initial incentive of selecting Well1 and Well2. As the reviewer suggested, further explanation and more details of the two wells will be included in the revised manuscript.

2. The number of wells selected in this study
Why did the authors focus only two wells? I believe there are many wells those provide long period of ground water data. As the authors mentioned, the results presented in the manuscript are site-specific. In that case, it would be much better to explain why only two wells were focused and analyzed in the manuscript.
*Authors' response*:
The reasons why the authors only focus on the two specific wells are: firstly, the groundwater level monitoring records of these two wells are long (70 and 80 years for Well1 and Well2 respectively). Long-term records can provide adequate data for fitting the probability density function. In addition, a larger sample of data can make the estimated Hurst exponent more stable (Weron, 2002). Secondly, we do agree with the reviewer that other long periods of groundwater datasets exist. For example, a dataset of groundwater monitoring in Texas, which was mentioned in the manuscript, includes more than 250 wells. Other long records can also be found in this dataset. However, comparing to the two wells selected in the study, these long records have large percentage of missing

data, which make them difficult to be analyzed. Thirdly, the authors found these two wells (Well1 and Well2) are very representative, i.e., one of them falls in the Brownian motion domain and the scaling pattern fluctuates in the investigated time intervals, while the other one illustrates the heavy-tailed characteristics and shows persistent scaling pattern. Focusing on the analysis of Well1 and Well2 can provide a more detailed picture of the groundwater level fluctuations. Therefore, Well1 and Well2 are chosen in specific to be analyzed in the manuscript. The manuscript will be revised to add the reasons mentioned above.

Reference
Weron, R.: Estimating long-range dependence: finite sample properties and confidence intervals, Physica A: Statistical Mechanics and its Applications, 312, 285-299, 2002.

---

## Author Comment (AC2) · 24 Jul 2017

**Response to Comments of Anonymous Referee #2**

The authors thank the anonymous referee #2 for the valuable comments and suggestions. Responses to the issues raised by the Reviewer #2 are provided below in red color:

This study conducted fractal scaling analysis of groundwater level fluctuations in confined aquifer wells by means of Detrended fluctuation analysis (DFA), Multifractal detrended fluctuation analysis (MF-DFA), and Multiscale Multifractal Analysis (MMA), and by investigating the relationship between the stability index and the Hurst exponent. In my opinion, this study provides important knowledge on groundwater level fluctuations with sufficient novelty, and this paper was mostly written well. However, some parts, especially Introduction, need to be improved before publication.

The authors thank the Reviewer #2 for the positive comments.

1. P.2 L.8-18: The relationship between this paragraph and this study is not clear. Please add some more explanations or descriptions.

More explanations will be added as suggested.

2. P.2 L.25: (MF-DFA(Kantelhardt et al., 2002) -> (MF-DFA, Kantelhardt et al., 2002)

The correction will be made.

3. P.3 L.20-21: Please add a reference for the sentence: "stability index and the Hurst exponent are related under certain conditions."

A reference (Taqqu et al., 1997) will be added as suggested.

4. P3. L.13-21: I strongly recommend the authors to explicitly explain the differences between existing related studies and this study in order to emphasize the novelty of this study. In addition, please briefly describe the purpose of each analysis here, and explain why the authors used long-term groundwater level data in this study.

This paragraph will be modified to emphasize the differences between this study and existing related studies. A brief description of the purpose of each analysis (DFA, MF-DFA and MMA) will be added. The reason for the selection of long-term groundwater level data will be further explained.

5. Equation (1): Please define t.

$t$ denotes time. The definition will be added in the revised manuscript.

6. P.4 L.1-3: I recommend the authors to explain in this section why it is important to detect evolving nonstationarities in this study.

As the reviewer recommended, explanation of the importance of detecting evolving nonstationarities will be added to the revised manuscript.

7. 3 Data Analysis: How were the two wells selected? Why were only two wells used in this study?

The reasons why the authors only focus on the two specific wells are: firstly, the groundwater level monitoring records of these two wells are long (70 and 80 years for Well1 and Well2 respectively). Long-term records can provide adequate data for fitting the probability density function. In addition, a larger sample of data can make the estimated Hurst exponent more stable (Weron, 2002). Secondly, we do agree with the reviewer that other long periods of groundwater datasets exist. For example, the dataset of groundwater monitoring in Texas, which was mentioned in the manuscript, includes more than 250 wells. Other long records can also be found in this dataset. However, comparing to the two wells selected in the study, these long records have large percentage of missing data, which make them difficult to be used. Thirdly, the authors found these two wells (Well1 and Well2) are very representative, i.e., one of them falls in the Brownian motion domain and the scaling pattern fluctuates in the investigated time intervals, while the other one illustrates the heavy-tailed characteristics and shows persistent scaling pattern. Focusing on the analysis of Well1 and Well2 can provide a more detailed picture of the groundwater level fluctuations. Therefore, Well1 and Well2 are chosen in specific to be analyzed in the manuscript. The manuscript will be revised to add the reasons illustrated above.

8. Figures 10 and 11: What are the red dot lines in the normal probability plots?

The red lines in Figures 10 and 11 denote the theoretical normal probability. The data probability curve would lie on the straight line if the data are Gaussian distributed. The authors will add such information in the revised manuscript.

9. Conclusions: Please add the summary of the investigation on the relationship between the stability index and the Hurst exponent.

The summary of the investigation on the relationship between the stability index and the Hurst exponent will be added in the revised manuscript.

Reference
Taqqu, M. S., Willinger, W., and Sherman, R.: Proof of a fundamental result in self-similar traffic modeling, ACM SIGCOMM Computer Communication Review, 27, 5-23, 1997.
Weron, R.: Estimating long-range dependence: finite sample properties and confidence intervals, Physica A: Statistical Mechanics and its Applications, 312, 285-299, 2002.

---

## Author Response (AR1)

The authors appreciate the valuable comments and suggestions from the handling Editor. The authors' responses are provided in red below.

Dear Authors,

After assessing the responses to reviewer comments, I am happy to inform you that the paper can be published subject to minor revisions. Here are a few things that need to be addressed.

Thanks to the handling Editor for the positive comments.

1) Please provide in the introduction, a broader context for hydrologic sciences stating the benefits of better understanding the scaling properties of groundwater fluctuations. For example, understanding these mechanisms can help investigate whether the features reflect in the existing groundwater, hydrologic and regional land atmosphere coupled models; hence providing an impetus for their enhancement.

The authors modified the introduction to better emphasize the importance of fractal scaling analysis of groundwater level fluctuations.

2) Furthermore, the response regarding the choice of the two wells is not convincing enough. Can you please provide (perhaps can be included in the supplemental material) a spatial distribution of other groundwater stations and their length records in the basin. At the very least, some analysis needs to be presented to show that these two wells are representative of the general groundwater flow properties in the basin. Perhaps, a simple dependency analysis with other stations in the basin could help demonstrate that. The exponents can vary systematically in the basin depending on where it is a homogenous or heterogeneous profile throughout. Regarding this, can you provide some more discussion on regionalization of these properties?

Thank you for the valuable comment. The authors added below section as Appendix A:

[revised manuscript text omitted]